# A Retrospective Analysis of High Resolution Ultrasound Evaluation of the “Split Fat Sign” in Peripheral Nerve Sheath Tumors

**DOI:** 10.3390/healthcare11243147

**Published:** 2023-12-12

**Authors:** Jeena B. Deka, Ritu Shah, Miguel Jiménez, Nidhi Bhatnagar, Alfredo Bravo-Sánchez, Inés Piñas-Bonilla, Javier Abián-Vicén, Fernando Jiménez

**Affiliations:** 1Dispur Polyclinic and Hospitals Pvt. Ltd., Guwahati 781006, India; jeena.bordoloideka@gmail.com; 2Faculty of Health Sciences, San Antonio Catholic University, 30107 Murcia, Spain; murcia.nidhi.bhatnagar@maxhealthcare.com (N.B.); josefernando.jimenez@uclm.es (F.J.); 3Seth GS. Medical College and King Edward Memorial Hospital, Mumbai 400012, India; rsritushah9@gmail.com; 4Asepeyo Hospital, 28823 Madrid, Spain; 5Radiology Department, Mata Devi Hospital Max Hospital, Panchsheel, New Delhi 110058, India; 6Facultad de Ciencias de la Salud, Universidad Francisco de Vitoria, 28223 Pozuelo de Alarcón, Spain; 7Faculty of Medicine, University of Extremadura, 06006 Badajoz, Spain; inespibonilla@gmail.com; 8Performance and Sport Rehabilitation Laboratory (DEPORSALUD), Faculty of Sport Sciences, University of Castilla-La Mancha, 13001 Toledo, Spain

**Keywords:** peripheral nerve sheath tumors, fatty tissue, schwannomas, neurofibromas, high resolution ultrasound

## Abstract

Peripheral nerve sheath tumors (PNST) comprise schwannomas and neurofibromas. The finding of increased adipose tissue around benign PNSTs has been described as the “split fat sign” on magnetic resonance imaging exams, which is suggestive of an intramuscular or intermuscular location of the tumor. However, few studies have described this sign as a salient ultrasound feature of PNSTs. The main purpose of this study was to retrospectively evaluate the presence of increased fatty tissue deposition around benign PNSTs diagnosed by high-resolution ultrasound. In addition, we aimed to corroborate the presence of vascularization around the affected area. A retrospective analysis of ten cases of PNSTs and two cases of post-traumatic neuromas diagnosed by high-resolution ultrasound was performed with a Logiq^®^ P8 ultrasound with a 2–11 MHz multifrequency linear probe L3-12-D (central frequency: 10 MHz). Localized types of neurofibromas and schwannomas in any location were seen as predominantly hypoechoic tumors with an oval or fusiform shape. Exiting and entering nerves (tail sign) were observed in six cases, showing localized lesions both in intermuscular and subcutaneous locations. The presence of increased hyperechoic tissue (the split fat sign) was noted in cases of solitary intermuscular and intramuscular peripheral nerve sheath tumors, mainly the schwannomas. Though small tumors did not demonstrate the tail sign, the increase in adipose tissue and vascularity on US was well demonstrated. In conclusion, the nerve in continuity forms the basis of the ultrasonographic diagnosis of PNSTs. However, high-resolution US can convincingly demonstrate the increased presence of fat in the upper and lower poles as well as circumferentially in intermuscular or intramuscular benign PNSTs.

## 1. Introduction

Schwann cells play a vital role in impulse transmission in the peripheral nervous system. They are a type of nervous tissue glial cell that performs functions of neuronal support and control of embryonic axon growth [1]. In addition to forming the myelin sheath around the axons of the peripheral nervous system, these cells are the only ones that stimulate the growth and regeneration of peripheral nerves [1]. Although the nature and origin of the factors involved in the proliferation processes are unknown, they develop in the embryonic process, after nerve injury, and in Schwann cell tumors [2].

Most tumors arising in the peripheral nervous system derive from Schwann cells or their precursors [3]. These neoplasms, which are commonly referred to as peripheral nerve sheath tumors (PNSTs), including neurofibromas, schwannomas, and malignant peripheral nerve sheath tumors, have a common neural origin [4], but intercellular interactions in the tumor’s microenvironment can produce tumors of different cell lineages [5]. They are often benign, but they can cause nerve damage and loss of function in the affected area [6].

Schwannomas are a type of benign tumor derived from the Schwann cells of the peripheral nerves. They are usually solitary and sporadic, although they can be seen in neurofibromatosis (NF) as multiple forms [7]. Their clinical presentation varies considerably; they can cause dysfunction due to a mass effect in the nerves where they are located. Meanwhile, the term neuroma is often used for post-traumatic swelling of nerves. They develop as an injured nerve begins to heal in an unregulated manner, resulting in a lump of disorganized axon fibers and non-neural tissue growth [8]. In this respect, it is important to establish a correct and differential diagnosis between them. The techniques of magnetic resonance imaging (MRI) and ultrasound (US) are the most commonly used diagnostic methods [9].

The differential diagnosis includes nerve lesions, such as ganglions, which have a cystic aspect, with posterior enhancement on ultrasound [10]. Nerve sheath tumors (schwannomas and neurofibromas) appear in the absence of previous trauma or in association with NF; on ultrasound, they have a fusiform morphology and are well-defined and hypoechoic. On MRI, they are iso or hypointense on T1, heterogeneous, and hyperintense on T2, and are enhanced after the administration of gadolinium [11]. Post-traumatic bulbs appear on ultrasound as a swelling of the nerve that can adopt the morphology of an oval hypoechoic pseudo-mass with loss of the normal fibrillar pattern of the nerve (comparison with the contralateral side assists diagnosis); the history of trauma together with the absence of the target sign is key to their differentiation [12].

In the last few years, the study of ultrasound signs, including the tail sign, split fat sign, and bull’s-eye sign, has become key to differential diagnosis. The tail sign, characterized by the divided arrangement of the fatty tissue due to the entrance or exit of the nerve, is the most typical feature [13]. It appears in more than 90% of PNSTs, with ultrasound showing a well-defined hypoechoic zone, eccentric in schwannomas, and central in neurofibromas [14]. Many studies have demonstrated that the appearance of this sign is crucial for diagnosing schwannomas [15]. The target sign appears in benign tumors where a well-demarcated hypoechoic lesion is observed and may show distal acoustic enhancement, simulating a cyst. A target signal can be seen in neurofibromas with a hypoechoic peripheral zone and a hyperechoic central zone [16]. The split fat sign, described as increased fat deposition in the upper and lower poles of peripheral nerve sheath tumors, is a commonly described MRI finding; however, few studies have mentioned it as an important ultrasound finding [14,17]. A hyperechogenic rim separates the fatty layer from the muscular layer. The echogenicity of the fatty tissue and its vascularization are studied using high-resolution ultrasound, which allows a differential diagnostic correction to be made [18].

Ultrasound has increasingly gained importance in recent years as a reliable and useful tool for evaluating peripheral nerve tumors. It enables us to evaluate tumors based on their size, location, encapsulation, echogenicity, cystic changes, presence of a target sign, and the presence of an existing or entering nerve. In addition, color Doppler imaging helps to determine the presence and amount of vascularization in a non-invasive way [19]. The main purpose of this study was to retrospectively evaluate the presence of increased deposition of fatty tissue, i.e., the split fat sign, around benign PNSTs diagnosed by high-resolution ultrasound. In addition, we aimed to corroborate the presence of vascularization around the affected area.

## 2. Materials and Methods

### 2.1. Participants

A retrospective study was carried out on 11 adults and a young girl diagnosed with PNST, NF, or traumatic neuromas, which were evaluated with high-resolution US. A clinical history was taken for each patient, documenting the age (range: 13 to 57 years), duration of symptoms (range: 1 to >25 years), complaint of any palpable swelling, and neurological manifestations. The preliminary questionnaire focused on relevant information pertaining to the presence of weakness, numbness, loss of sensation, and tingling. A history of any injury or surgical intervention was also taken. Patients with other disorders were not included in the study. The demographic data are shown in Table 1.

The Ethics Committee of Clinical Research at the Dispur Polyclinic and Hospitals Pvt Ltd. Guwahati, Guwahati, India, approved this study. In addition, written informed consent was obtained from all participating patients.

### 2.2. Clinical Characteristics

Ten cases of PNST, examined by high-resolution US, were evaluated retrospectively to examine the presence of increased adipose tissue around the tumors. Five cases were solitary Schwannomas in the lower extremity (4 were intermuscular and 1 was intramuscular), and 2 cases of solitary neurofibromas were detected in the forearm and hip region (1 was intermuscular and 1 was subcutaneous). Besides, 3 cases of neurofibromas associated with neurofibromatosis were evaluated (1 was the diffuse variety, and 2 were localized) in the head, neck, and trunk regions.

Finally, 2 cases of post-traumatic neuromas, which are mimics of peripheral nerve sheath tumors, were also evaluated. One was a case of complete nerve transection of the ulnar nerve of the forearm, resulting in proximal and distal stump neuromas; the other case was a neuroma in continuity. Both were intermuscular and located in the forearm.

The diagnosis was confirmed surgically and by histopathological examination (HPE) in 8 of the 12 cases. Two cases of neurofibromatosis were not operated on, while 2 cases of a large PNST from a major nerve trunk were lost to follow-up. MRI correlations were not carried out in any of the cases.

### 2.3. US Examination

All patients underwent routine 2D B-mode ultrasound examination. All US exams were carried out by an expert with more than ten years of experience in nerve ultrasound. The US exams were performed with a Logiq^®^ P8 ultrasound (GE Healthcare, Milwaukee, WI, USA) with a 2–11 MHz multifrequency linear probe L3-12-D (GE Healthcare system, Milwaukee, WI, USA). The frequency, focus, and depth were adjusted in each case (central frequency: 10 MHz), and the probe and its angle of insonation were kept perpendicular to the nerve to prevent anisotropy. The patients were placed in a comfortable position for the examination of peripheral nerves in the limbs. The examination was always started from a known anatomical landmark, and the nerve was kept in the center of the image during the movement of the transducer along the entire length of the nerve, from the proximal to distal side. The examinations were performed on short and long axes, and the nerves were identified by their typical honeycomb appearance. In addition, sonopalpation was applied to try to reproduce the patient’s symptoms. A dynamic study was also conducted to observe the sliding of nerves in relation to surrounding muscles and soft tissues.

The upper limb nerves were evaluated with the patient lying on their back or in a sitting position facing the radiologist. The ulnar nerve was first identified in the elbow behind the medial epicondyle. The median nerve was examined by first identifying it medial to the brachial artery in the anterior elbow. The radial nerve was first identified in the elbow between the brachialis and brachioradialis muscles. The lower limb nerves were identified when the patient was lying down in a supine or prone position, depending on the nerve under study. The tibial nerve was first identified in the popliteal fossa with the patient prone or in the tarsal tunnel with the patient supine. In the 3 cases of neurofibromatosis type 1, cutaneous neurofibromas in the neck and trunk region, the lesions were directly evaluated using a higher frequency probe and settings. The localized neurofibromas in the trunk were evaluated for nerve continuity, target sign, and vascularity. The internal architecture, deeper extent, and vascularity of the diffuse variety in the neck were documented.

Mass-like enlargements were further evaluated by studying the echogenicity, shape, fascicular pattern, and internal architecture, such as calcification, cysts, or hyperechoic areas suggestive of the target sign. In the short axis, the nerve cross-sectional area (CSA) changes and discontinuity of the nerve were noted. In the long axis, the normal nerves reveal a bundle with a straw appearance due to the regular arrangement of fascicles. The shape and size of the masses were documented. The main focus was also to demonstrate the continuity of the nerves with the masses proximally and distally to confirm their neural origin and document whether the masses were central or eccentrically positioned in relation to the nerves. A power Doppler exam was employed to analyze the internal vascularity of the solid component. The symptoms of each of the neuromas were recorded independently in each individual; however, the results were presented collectively after the analysis.

## 3. Results

The summary of the main outcomes is shown in Table 2. Localized types of neurofibromas and schwannomas in any location were seen as predominantly hypoechoic tumors with an oval or fusiform shape.

Exiting and entering nerves (tail sign) were observed in six cases, showing localized lesions in intermuscular and subcutaneous locations. They were easily observed in those arising from major nerve trunks (Figure 1A). In the case of smaller nerves, higher resolution probes and settings helped demonstrate exiting and entering nerves with greater confidence (Figure 2). In three small tumors, the tail sign could not be demonstrated, but subsequent surgery revealed exiting and entering nerves (Figure 3). Cystic changes were detected in three cases of schwannomas. A large schwannoma from the tibial nerve showed cysts with blood fluid levels and significant internal vascularity (Figure 1B).

Stump neuromas revealed a hypoechoic bulbous mass-like enlargement of the proximal and distal ends of the transected nerve (Figure 4). A well-defined fusiform mass at the site of transection with nerve continuity proximally and distally was observed in the neuroma in continuity (Figure 5). The tail sign was noted in both.

The “target sign” described as a hypoechoic lesion with a hyperechoic center was seen in three cases of subcutaneous neurofibromas both solitary and associated with NF1 (Figure 2, Figure 6 and Figure 7). It was not observed in the traumatic neuromas or schwannomas.

A diffuse variety of neurofibroma in a case of neurofibromatosis 1 showed hypoechoic subcutaneous thickening with hyperechoic spots and a reticular pattern. With color Doppler, mild vascularity was seen (Figure 8); the tail sign was absent.

The presence of increased hyperechoic adipose tissue, i.e., the split fat sign, which is the main focus of our retrospective analysis, was noted in cases of solitary intermuscular and intramuscular peripheral nerve sheath tumors, mainly the schwannomas (90% of cases examined). Though small tumors did not demonstrate the tail sign, the increase in adipose tissue and vascularity on US was well demonstrated. They were later confirmed operatively and histopathologically (Figure 2).

Interestingly, this increased hyper-echogenicity was also seen in relation to the stump neuromas (Figure 3). The neuroma in continuity also showed a hyperechoic rim of fat, which was more prominent in one end (Figure 4). Increased adiposity was not seen in the localized or diffuse sub-cutaneous neurofibromas (Figure 2, Figure 6, Figure 7 and Figure 8). There was an overlap in US findings between neuromas and Schwannomas in our case series. The presence of the split fat sign on high-resolution US further helped make a convincing diagnosis, especially in the absence of the tail sign in very small tumors. Surgery confirmed the neural origin of the tumors by demonstrating the nerve continuity. Excision biopsy and HPE provided the definitive diagnosis of the tumor type.

## 4. Discussion

The main purpose of this study was to determine if the presence of increased fatty tissue deposition, i.e., the split fat sign around benign PNSTs diagnosed by high-resolution ultrasound, is confirmed after surgery, adding a new ultrasound application in neuroma and schwannoma diagnosis. In addition, we aimed to corroborate the presence of vascularization around the affected area. Our results showed that in up to 90% of cases, ultrasound was able to detect fat deposition. PNSTs are neoplasms arising from a peripheral nerve or that show nerve sheath differentiation that grow along the long axis of a nerve [17]. They can be benign or malignant, comprising 12% of all benign and 8% of all malignant soft tissue neoplasms affecting all age groups [17,20]. Benign PNSTs are schwannomas and neurofibromas [19,20], while malignant PNST is a general term used to describe malignant entities of both types as the cell of origin is often unknown. PNSTs can affect any peripheral nerve and are most commonly intermuscular, followed by subcutaneous locations. Intramuscular PNSTs are the least frequent [21].

Neurofibromas are further classified into localized, diffuse, and plexiform [20]. The localized type usually arises from the cutaneous nerves, the diffuse type originates from the nerves in the subcutaneous tissue of the head and neck, while the plexiform type is a diffuse type that runs tortuously along the branches of the main nerve. Traumatic neuromas are pseudo-tumors and mimics of PNSTs, and are related to disorganized non neoplastic proliferation of axonal tissue in response to a partial or completely transected nerve [12,17]. Both benign and malignant PNSTs can be associated with neurofibromatosis [20]. MRI and US are the best modalities to characterize nerve lesions [17]. In our research, the increased hyper-echogenicity suggestive of increased deposition of adipose tissue is an important ultrasound finding seen around benign PNSTs in the intermuscular and intramuscular location, but not in the subcutaneous location, agreeing with previous research [15,22].

The nerve in continuity with a mass showing entering and exiting nerves, also called the “tail sign”, is the most reliable feature of PNSTs evaluated with MRI and US techniques [14,17,23]. On US, the mass is well-defined, hypoechoic, fusiform, or round with a hyperechoic capsule and oriented along the long axis of a nerve [21,24]. In our study, the “tail sign” was described in four of five cases, and hence, it followed the common findings of US exploration. The location in relation to the nerve is eccentric in schwannomas and central in neurofibromas [18,23,25]. Several features of PNSTs have been described in both modalities, with considerable overlap between them [13,18,19,25]. Three signs described as frequent MRI findings in these types of tumors will be included in this discussion, along with an attempt to compare them with the US characteristics described in the literature. It is often difficult to distinguish between schwannomas and neurofibromas by sonography [14]. These signs can also be seen in stump neuromas and neuromas in continuity present in the intermuscular location.

The tail sign on US appears in more than 90% of PNSTs [14,21]. In our case, it was observed in ~80% of the cases. This could be due to the difficulty in finding an entering or exiting nerve, as the peripheral nerve branch is either small or easily confused with adjacent hypoechoic fat and connective tissue [19]. The nerve must also be scanned on the short axis as compressed fascial planes may mimic a nerve on the long axis [21]. Vascularity and cystic areas due to myxoid degeneration, hemorrhage, or necrosis are seen more in long-standing ancient schwannomas [23,25]. The “target sign” is observed on MRI and US, more so in neurofibromas and rarely in schwannomas. In agreement with this, the “target sign” was only observed in neurofibromas. On US it represents a hyperechoic center with a hypoechoic rim due to myxoid tissue surrounding a central area of fibrous tissue [21,25]. The “split fat sign” is a known MRI feature of benign intermuscular and intramuscular PNSTs but is not specific to them [13,26]. It represents fat deposition seen as a tapered rim of fat signal adjacent to the proximal and distal ends of a lesion. It is due to the remodeling of fat around the neuromuscular bundle as the tumor grows in the intermuscular space [21,27,28].

The cause of fat in intramuscular PNSTs is unclear [12,13]. Fatty rinds seen at both ends or circumferentially, separating tumors from surrounding muscle, have been described on MRI [29]. They are reported more in benign tumors and are thought to be caused by a displaced neurovascular bundle, fat, and fatty atrophy [26]. In malignant PNSTs, the rim of fat is incomplete or obliterated due to an infiltrative growth pattern [20]. The “split fat sign” or fatty rinds caused by the increased fat deposition have not been frequently described as a salient ultrasound finding in PNSTs in most of the literature. Tagliaco et al. [17] however mentioned that the split fat sign may be seen on US as a peripheral rim of fat in PNSTs affecting large nerves, coinciding with our results.

Schwannomas may show a hyperechogenic rim of fat separating the mass from the surrounding muscle, thought to be due to a fibrous capsule, fibrous pseudo-capsule, or compression of the surrounding fat [21]. Our review showed significant hyper-echogenicity around the intermuscular and intramuscular PNSTs, especially in the upper and lower poles, which is suggestive of the split fat sign. They were not seen in the cutaneous or diffuse variety of neurofibromas.

The US features of diffuse neurofibromas are the thickening of the skin and subcutaneous tissue with a reticular hypoechoic pattern without a definite discrete mass, although hyperemia is noted [25]. Plexiform neurofibromas are diffuse nodular enlargements of nerves presenting as a discrete lobulated mass or lobulated neural enlargement with a “bag of worms” appearance.

Traumatic neuromas in continuity are seen as focal hypoechoic fusiform or disorganized masses with a normally appearing entering or exiting nerve at the transection site. Stump neuromas are hypoechoic bulbous masses in continuity with a nerve. There is no mention of the split fat sign as an MRI or US finding [17,30]. The two cases of intermuscular traumatic neuromas in our analysis interestingly demonstrated increased surrounding fat on US. This increased adipose tissue deposition was an important US finding around benign PNSTs situated in the intermuscular and intra-muscular locations but not in the subcutaneous location, agreeing with previous research [15].

Finally, this research allows us to define the diagnostic criteria for peripheral nerve sheath tumors with ultrasound. An economical, innocuous, and reproducible tool that offers a promising perspective in the diagnosis of these tumors. Research on their radiological criteria is key for diagnosis, pre-treatment study, disease prognosis, and evaluating possible remission of neural tumors. Our study opens up new possibilities in this line of research, which could be very useful for advancing the study of these diseases.

Our study has some limitations. First, the sample size was limited, and the heterogeneity of the cases analyzed did not allow for comparisons between groups. Furthermore, we did not perform a reproducibility study of the findings. However, to avoid the evaluator effect, all examinations were performed by the same expert with more than ten years of experience. Finally, it would have been interesting to add a comparison with other diagnostic techniques, such as MRI, to establish correlations prior to surgery.

## 5. Conclusions

Nerve continuity forms the basis of the ultrasonographic diagnosis of PNSTs. However, high-resolution US can convincingly demonstrate the increased presence of fat in the upper and lower poles as well as circumferentially in intermuscular or intramuscular benign PNSTs. Described more often on MRI as the “split fat sign” and fatty rinds, we believe this can also be a salient high-resolution ultrasound feature of PNSTs. Importantly, it can be a clue towards the diagnosis, especially in inter- and intramuscular PNSTs of small nerves, where the entering or exiting nerve or the nerve in continuity may not be demonstrated. The increased adiposity is not seen in the cutaneous, diffuse, or plexiform variety of neurofibromas.

The finding of increased adiposity may also be seen in intermuscular traumatic neuromas, which warrants further research with more case studies in the future.

Key points: (1) Ultrasound examination can diagnose peripheral nerve sheath tumors and determine their location based on the presence of fatty tissue near the lesion. (2) Ultrasound diagnosis of benign PNSTs established that in 90% of cases, adipose tissue is present near intermuscular or intramuscular lesions. (3) The use of ultrasound as a diagnostic tool can reduce the economic and time cost of consultations.

## Figures and Tables

**Figure 1 healthcare-11-03147-f001:**
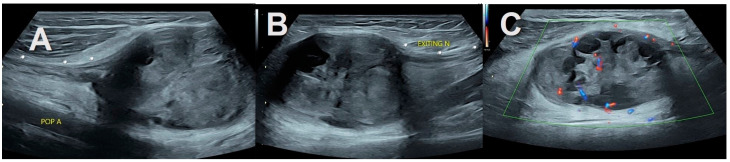
Schwannoma from posterior tibial nerve. A 57-year-old man presented with dull pain in the posterior knee for the previous 8 months. (**A**)—Long axis view in popliteal fossa reveals a well-defined mass with internal cystic components and blood fluid levels within it. (**B**)—An entering and exiting nerve are pathognomonic of a nerve tumor. Increased fatty tissue is seen at the upper and lower poles of the mass and surrounding it. (**C**)—Power doppler correlation reveals internal vascularity of the solid component.

**Figure 2 healthcare-11-03147-f002:**
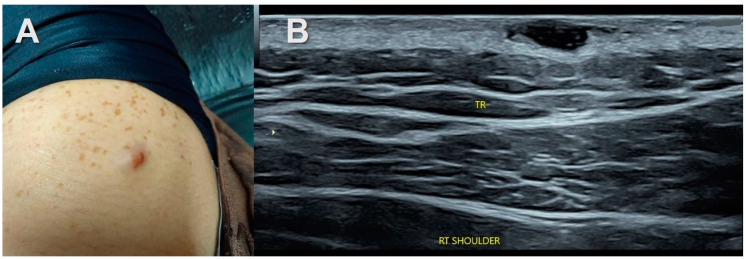
Neurofibromatosis type 1. (**A**)—25-year-old woman with neurofibromatosis type 1 showing café-au-lait (CAL) spots with a few small reddish bumps in the shoulder region. (**B**)—Gray scale ultrasound shows multiple discreet hypoechoic oval lesions in the subcutaneous region of the shoulder. Entering and exiting nerves were seen. No increased adipose tissue was observed.

**Figure 3 healthcare-11-03147-f003:**
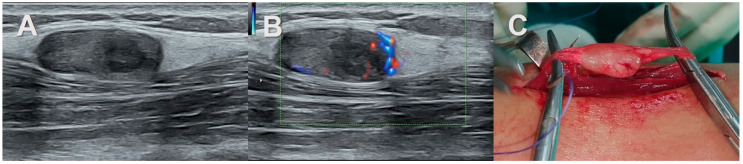
Intramuscular schwannoma anterior leg. A 25-year-old woman presented with pain in the left upper anterior leg for the previous 4 years. A small palpable swelling was felt which was painful on pressure. (**A**)—well defined hypoechoic oval mass with a small anechoic area within the extensor digitorum longus just deep to the epimysium. Increased fatty component surrounding the mass mainly in the upper and lower poles. No definite entering or exiting nerve was seen. (**B**)—Power Doppler shows vascularity within it. (**C**)—Per-operative images showed a mass with entering and exiting nerves. A histopathological examination (HPE) correlation revealed a schwannoma.

**Figure 4 healthcare-11-03147-f004:**
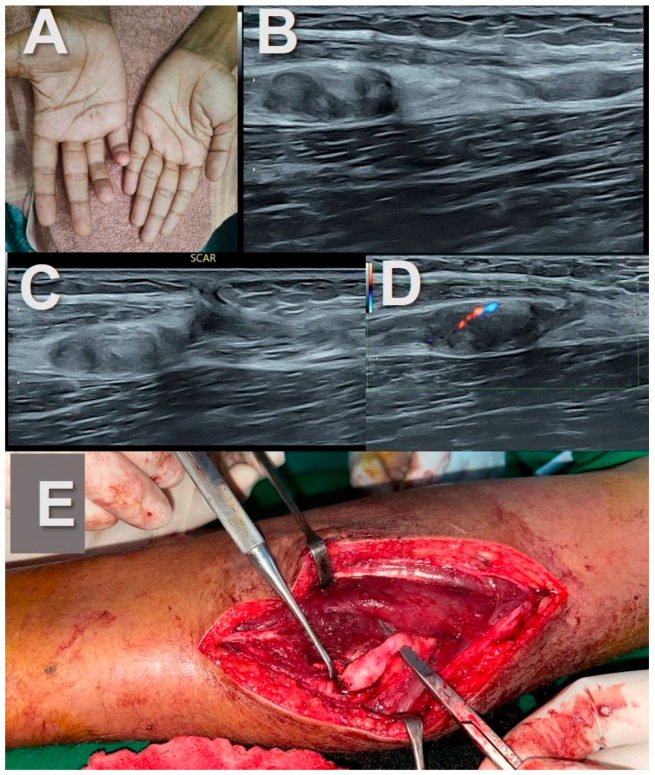
Stump neuromas of the ulnar nerve. (**A**)—A young man with clawing of the right 4th and 5th digits following lacerating injury in mid forearm along with fracture of both bones in forearm one year before. (**B**,**C**)—Ultrasound image reveals complete transection of the ulnar nerve at site of scar with mass-like enlargement of the proximal and distal ends of the nerve suggestive of stump neuromas. Increased adipose tissue is also noted at the site of neuroma formation in the intermuscular location and the gap between them. (**D**)—Mild vascularity is noted on power doppler. (**E**)—Intra-operative picture of the stump neuroma.

**Figure 5 healthcare-11-03147-f005:**
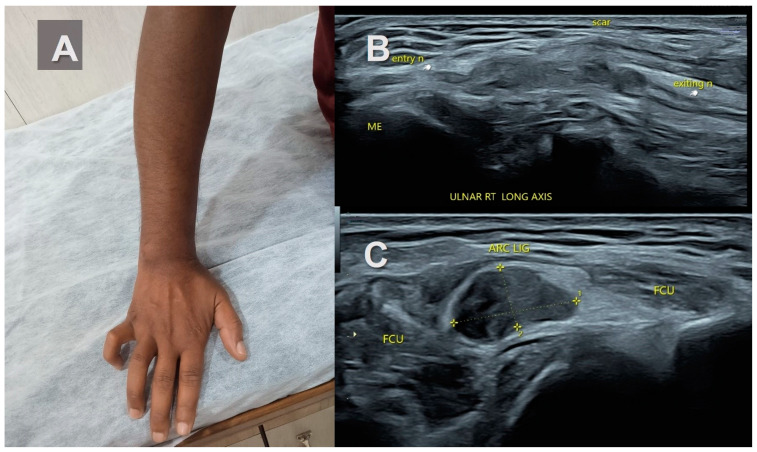
Neuroma in continuity of the ulnar nerve. (**A**)—A 29-year-old man with cut injury in right forearm from 1 year before. Clawing of the 4th and 5th digits. (**B**,**C**)—B mode ultrasound in long and short axis: A 16 × 6 mm-sized well defined mass was noted in the course of the ulnar nerve at the level of the true cubital tunnel deep to the arcuate ligament (ARC LIG) and between the two heads of the flexor carpi ulnaris muscles (FCU). It was continuous with the ulnar nerve proximally and distally suggestive of neuroma in continuity. Increased adipose tissue was noted surrounding the mass. Scar is evident in the superficial soft tissues.

**Figure 6 healthcare-11-03147-f006:**
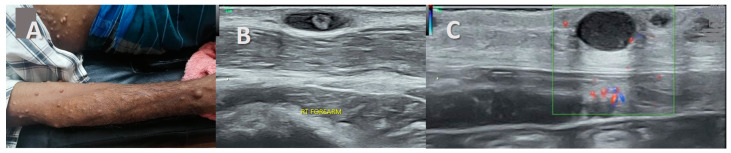
Localized neuromas in neurofibromatosis type 1. (**A**)—A 46-year-old man with neurofibromatosis type 1 and neuromas involving head, neck, trunk, and limbs. (**B**)—Ultrasound image shows multiple discreet hypoechoic oval lesions in the subcutaneous region of the forearm. The target sign “described as hypoechoic lesion with a hyperechoic center” is visible. (**C**)—Power Doppler shows vascularity.

**Figure 7 healthcare-11-03147-f007:**
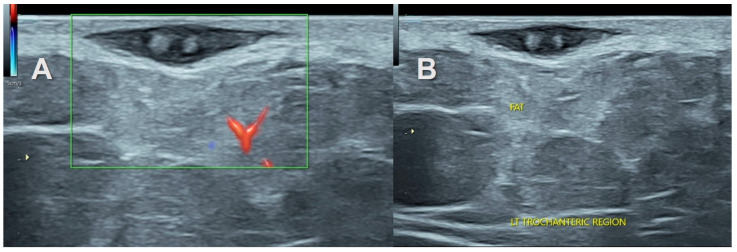
Localized subcutaneous neurofibroma in a 34-year-old man with palpable swelling in the left lateral hip region. (**A**)—Gray scale ultrasound with color Doppler shows focal hypoechoic oval lesion in the subcutaneous trochanteric region. (**B**)—The target sign as well as entering and exiting nerves are visible.

**Figure 8 healthcare-11-03147-f008:**
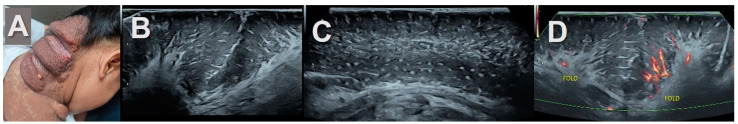
(**A**)—Neurofibromatosis type 1 with diffuse neurofibromas and CAL spots in the nape of the neck in a 13-year-old girl. (**B**,**C**)—Long and short axis gray scale ultrasound shows hypoechoic thickening of skin and subcutaneous tissue with a reticular and spotty hyperechoic pattern. There is no definite discrete mass or increase in adiposity. (**D**)—Mild hyperemia is noted.

**Table 1 healthcare-11-03147-t001:** Summary of the demographic data.

Patient	Age (Years)	Sex	Time Evolution (Years)	CAL Presence	Familiar Presence
Case 1	9	Female	9	Yes	Yes
Case 2	45	Male	30	No	Yes
Case 3	38	Female	9	Yes	Yes
Case 4	28	Female	9	Yes	-
Case 5	25	Female	10	Yes	-
Case 6	30	Male	1	No	-
Case 7	32	Male	3	No	-
Case 7	29	Male	0.5	No	-
Case 8	16	Female	4	No	-
Case 9	57	Male	1	No	-
Case 10	40	Male	2	No	-
Case 11	27	Male	1	No	-

CAL: café-au-lait spots presence.

**Table 2 healthcare-11-03147-t002:** Summary of results of ultrasound examination.

Nerve Tumor	N	Tail Sign	Target Sign	Increased Adiposity
1. PNST	10			
Schwannoma	5			
Solitary	5	+ (4 out of 5)	−	+
Intermuscular	4	+	−	+
Intramuscular	1	−	−	+
Neurofibromas	5			
Solitary	2	+	+	+ (1 out of 2)
Intermuscular	1	+	+	+
Subcutaneous	1	+	+	−
NF 1 Subcutaneous	3	+ (2 out of 3)	+ (2 out of 3)	−
Localized	2	+	+	−
Diffuse	1	−	−	−
2. PTNI	2	+	−	+

PNST = Peripheral nerve sheath tumors; PTNI = Post-traumatic neuromas—Intermuscular; NF = Neurofibromatosis; + = found; − = not found.

## Data Availability

Data supporting this study are available from the corresponding author on request.

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
