# Peer review of "A Retrospective Analysis of High Resolution Ultrasound Evaluation of the “Split Fat Sign” in Peripheral Nerve Sheath Tumors"

_healthcare, 2023, doi:10.3390/healthcare11243147_

Round 1

Reviewer 1 Report

Comments and Suggestions for Authors

Authors proposed an Ultrasound evaluation of peripheral nerve sheath tumors. I have the following comments. 

1. The major contributions of this study should be listed clearly in the introduction. 

2. The motivation is not very clear. 

3. Comparison with previous studies and other imaging modalities such as magnetic resonance imaging should be done. 

4. Study sample is too small and should contain wider patients. 

5. What is the diagnostic accuracy of your method for identifying the peripheral nerve sheath tumor? 

6. Can this be automated?

Comments on the Quality of English Language

Minor corrections are required. 

Author Response

The authors would like to thank the Reviewer for their helpful comments and suggestions and for their assistance in improving the quality of the manuscript. The comments are addressed individually below. Changes within the manuscript are highlighted (in red). We believe that our manuscript is improved in the light of the suggested changes.

Authors proposed an Ultrasound evaluation of peripheral nerve sheath tumors. I have the following comments. 

  1. The major contributions of this study should be listed clearly in the introduction. 

Thank you for the comment. The major contributions had been listed in the key points section, although we considered better to include it at the end of the conclusions section. Now it reads: “Key points: (1) Ultrasound examination can be used to diagnose peripheral nerve sheath tumors and determine their location based on the presence of fatty tissue near the lesion. (2) Ultrasound diagnosis of benign PNSTs established that in 90% of cases there is the presence of adipose tissue near the lesions that are intermuscular or intramuscular in location. (3) The use of ultrasound as a diagnostic tool can reduce the economic and time cost of consultations”.

  1. The motivation is not very clear. 

Thank you for your comment. Our stress was that the split fat sign can also be an additional ultrasound sign which is hardly described in the literature. It could really help in making a more convincing diagnosis. It has been highlighted in the objective of the manuscript and now it reads: “The main purpose of this study was to retrospectively evaluate the presence of an increased deposition of fatty tissue i.e., the split fat sign around benign PNSTs diagnosed by high resolution ultrasound. In addition, we also aimed to corroborate the presence of vascularization around the affected area.

  1. Comparison with previous studies and other imaging modalities such as magnetic resonance imaging should be done. 

Done.

  1. Study sample is too small and should contain wider patients. 

Thank you for your comment. Although you consider that the number of participants was small, the heterogeneity of the sample made it relevant to know the main findings of different neuromas through ultrasound examination.

  1. What is the diagnostic accuracy of your method for identifying the peripheral nerve sheath tumor? 

Thank you for your comment. If the reference is to those cases where the tail sign or the target sign was observed, the fat accumulation sign was observed in 90% of the cases after ultrasound examination. Therefore, it seems that the split fat sign could be taken as a further diagnostic indicator on US examination. This percentage has been included in the results and discussion part of the manuscript. Now it reads: “The presence of increased hyperechoic tissue i.e., the split fat sign which is the main focus of our retrospective analysis, was noted in cases of solitary intermuscular and intramuscular peripheral nerve sheath tumors, mainly the schwannomas (90% of cases examined).

  1. Can this be automated? – Please, give us a feedback.

Thank you for your comment. All ultrasound (US) diagnoses were done by an expert with many years’ experience, so experienced evaluators could include these signs in future clinical exams and improve the neuroma examination. The split fat sign can be a very important feature in identifying small tumors, so that split fat sign could also be an additional US sign which is hardly described in literature. It can really help in making a more convincing diagnosis. This affirmation was included in the discussion section.

Reviewer 2 Report

Comments and Suggestions for Authors

The manuscript is interesting and has practical value, but needs minor correction.

1.     Page 4 line 183: I did not see the interpretation of CAL in the text. Specify please. 

2.     Page 5 line 193: Specify please “HPE correlation”

3.     Figure 4 describes a case of stump neuromas of the ulnar nerve. What surgical operation was performed in this case? Excision or excision+ reconstruction? (the purpose of the question is not to correct  the manuscript)

4.     In figure 5 (B,C), indicate the anatomical elements on the figure listed in legend: cubital tunnel, arcuate ligament, flexor carpi ulnaris muscle. Accordingly, an explanation will be added to the figure legend.

5.     The authors note that in eight cases the diagnosis was confirmed surgically and by histological examination. If the diagnoses overlapped in each case, note this in the results.

6.     Can the ultrasound evaluation of the “split fat sign” be successfully used in the diagnosis of cervical vagal schwannoma? (the purpose of the question is not to correct  the manuscript)

Author Response

The authors would like to thank the Reviewer for their helpful comments and suggestions and for their assistance in improving the quality of the manuscript. The comments are addressed individually below. Changes within the manuscript are highlighted (in red). We believe that our manuscript is improved in the light of the suggested changes.

The manuscript is interesting and has practical value, but needs minor correction.

Thank you for your comment.

  1. Page 4 line 183: I did not see the interpretation of CAL in the text. Specify please.

Done. The interpretation of CAL (café-au-lait) has been included in the manuscript.

  1. Page 5 line 193: Specify please “HPE correlation”

Done. Now it reads: “A histopathological examination (HPE) correlation revealed a schwannoma

  1. Figure 4 describes a case of stump neuromas of the ulnar nerve. What surgical operation was performed in this case? Excision or excision+ reconstruction? (the purpose of the question is not to correct the manuscript)

Thank you for your comment. The intervention was excision with sural nerve graft.

  1. In figure 5 (B,C), indicate the anatomical elements on the figure listed in legend: cubital tunnel, arcuate ligament, flexor carpi ulnaris muscle. Accordingly, an explanation will be added to the figure legend.

Thank you for the comment. The figure legend has been modified to include an explanation of this elements. Now it reads “Figure 5. Neuroma in continuity of the ulnar nerve. A -A 29-year-old man with cut injury in right forearm from 1 year before. Clawing of 4th and 5th digits. B and C - B mode ultrasound in long axis and short axis: A 16 x 6mm-sized well defined mass was noted in the course of the ulnar nerve at the level of the true cubital tunnel deep to the arcuate ligament (ARC LIG) and between the 2 heads of the flexor carpi ulnaris muscles (FCU). It was continuous with the ulnar nerve proximally and distally suggestive of neuroma in continuity. Increased adipose tissue was noted surrounding the mass. Scar was noted in the soft tissues superficial to it.

  1. The authors note that in eight cases the diagnosis was confirmed surgically and by histological examination. If the diagnoses overlapped in each case, note this in the results.

Thank you for your comment. At this moment it is not possible to definitely say if it is a Schwanomma or neurofibroma on Ultrasound (US) alone as there is overlap of findings. The main doctor diagnosed it as a peripheral nerve sheath tumor (PNST) based on the tail sign, but mainly the fat split sign was seen. If it was definitely eccentric, then the doctor diagnosed it as a PNST likely to be schwanomma or, if the target sign was seen then neurofibroma. In the small tumor on the leg the doctor could not see the tail sign and only the increased fat and tinel sign based on which he gave PNST as the probable diagnosis, but it was confirmed surgically and by histopathological examination (HPE). So, on US there was an overlap of findings between neuroma and Schwanomma. In our case series, US helped in making the diagnosis of PNST. Surgery confirmed the tumor with nerve continuity. Excision biopsy and HPE gave the definitive diagnosis of type of tumor. It has been included in the results section. Now it reads: “Interestingly this increased hyperechogenicity was also seen in relation to the stump neuromas (Figure 3). The neuroma in continuity also showed a hyperechoic rim of fat which was more prominent in one end (Figure 4). Increased adiposity was not seen in the localized or diffuse subcutaneous neurofibromas (Figures 2,6,7,8). There was an overlap of US findings between neuromas and Schwannomas in our case series. The presence of the split fat sign on high resolution US further helped in making a convincing diagnosis especially in the absence of the tail sign in very small tumors. Surgery confirmed the neural origin of the tumors by demonstrating the nerve continuity. Excision biopsy and HPE gave the definitive diagnosis of type of tumor.

  1. Can the ultrasound evaluation of the “split fat sign” be successfully used in the diagnosis of cervical vagal schwannoma? (the purpose of the question is not to correct the manuscript)

Thank you for your comment. Although you mention magnetic resonance imaging as the best method to describe the split fat sign in cervical schwannoma (Zeng et al., 2017) Reynolds et al. (2004) successfully described the split fat sign with ultrasound. Therefore, we considered that ultrasound could evaluate the split fat sign, although new studies are necessary to corroborate it.

Reynolds DL Jr, Jacobson JA, Inampudi P, Jamadar DA, Ebrahim FS, Hayes CW. Sonographic characteristics of peripheral nerve sheath tumors. AJR Am J Roentgenol. 2004 Mar;182(3):741-4. doi: 10.2214/ajr.182.3.1820741. PMID: 14975979.

Zheng X, Guo K, Wang H, Li D, Wu Y, Ji Q, Shen Q, Sun T, Xiang J, Zeng W, Chen Y, Wang Z. Extracranial schwannoma in the carotid space: A retrospective review of 91 cases. Head Neck. 2017 Jan;39(1):42-47. doi: 10.1002/hed.24523. Epub 2016 Jul 21. PMID: 27442804.

Reviewer 3 Report

Comments and Suggestions for Authors

The article is quite complex and sloppyly written. An original article must include statistical analysis; The relevant part is not available in this article.

The number of cases is insufficient. In addition to the sonographic sign mentioned in the title, other features were also investigated, but these were not specified in the title and purpose. The material and method section is careless and contains results information. There are no limitations in discussion.

Author Response

The authors would like to thank the Reviewer for their helpful comments and suggestions and for their assistance in improving the quality of the manuscript. The comments are addressed individually below. Changes within the manuscript are highlighted (in red). We believe that our manuscript is improved in the light of the suggested changes.

The article is quite complex and sloppyly written. An original article must include statistical analysis; The relevant part is not available in this article.

Thank you for your comment. Due to the heterogeneity of the sample, this article is a case series design, so the results are treated individually, without statistical treatment of the data. In future studies with a more homogeneous sample, which will allow us to divide the sample into groups and make comparisons, we will include the statistical analysis section.

The number of cases is insufficient. In addition to the sonographic sign mentioned in the title, other features were also investigated, but these were not specified in the title and purpose. The material and method section is careless and contains results information.

Thank you for your comment. Although you consider the number of participants was small, the heterogeneity of the sample made it relevant to know the main findings of different neuromas through ultrasound (US) examination. The most important finding of our study was that, despite the heterogeneity of the sample, 90% of the cases examined showed accumulation of adipose tissue adjacent to the tumor. Therefore, although after US examination there was an overlap of findings between neuroma and Schwanomma, US helped in making the diagnosis of PNST. Surgery confirmed the tumor with nerve continuity. Excision biopsy and a histopathological examination (HPE) gave the definitive diagnosis of type of tumor. All the investigated features have been highlighted in the purpose of the paper. Now it reads: “The main purpose of this study was to retrospectively evaluate the presence of an increased deposition of fatty tissue i.e., the split fat sign, around benign PNSTs diagnosed by high resolution ultrasound. In addition, we also aimed to corroborate the presence of vascularization around the affected area.

Finally, the clinical characteristics subsection in the material and methods section helps to describe the patients and their pathologies. Therefore, the authors considered that this information should be included in the material and methods section so that it is possible to have an overview of the cases examined.

There are no limitations in discussion.

Thank you for your comment. The limitations of the study have been included in the manuscript. Now it reads: “Our study has some limitations. First, the sample size was limited and the heterogeneity of the cases analyzed did not allow for comparisons between groups. Furthermore, we did not perform a reproducibility study of the findings, although to avoid the evaluator effect, all examinations were performed by the same expert with more than 10 years’ experience. Finally, it would have been interesting to add a comparison with other diagnostic techniques such as MRI, in order to establish correlations prior to surgery.”

Reviewer 4 Report

Comments and Suggestions for Authors

Dear authors, 

I read the study titled "A Retrospective Analysis of High Resolution Ultrasound Evaluation of the 'Split Fat Sign' in Peripheral Nerve Sheath Tumors" with great interest. The main goal of this study was to conduct a retrospective assessment of the presence of heightened accumulation of adipose tissue, specifically referred to as the split fat sign, surrounding benign peripheral nerve sheath tumors (PNSTs) as detected using high-resolution ultrasound.

The pictorial representation shows a remarkable visual appeal, showcasing a limited number of high-quality images. However, in order to meet the criteria for publishing, the article could benefit from further improvements.

1. Based on the term "split fat sign" and your assertion that this is novel approach, like "no study has described it as an ultrasound feature of PNSTs,", I believe this statement needs to be revised.  Several academic papers have been published that examine the alterations associated with adipose tissue growing, alongside the detection PNSTs using ultrasonic imaging. For example, Sonographic Characteristics of Peripheral Nerve Sheath Tumors - David L. Reynolds, Jr.Jon A. JacobsonPrasuna InampudiDavid A. JamadarFarhad S. Ebrahim, and Curtis W. Hayes, American Journal of Roentgenology 2004 182:3741-744 or Lawande AD, Warrier SS, Joshi MS. Role of ultrasound in evaluation of peripheral nerves. Indian J Radiol Imaging. 2014 Jul;24(3):254-8. doi: 10.4103/0971-3026.137037. PMID: 25114388; PMCID: PMC4126140.

2. From my perspective, the discussions reveal a degree of superficiality. It is recommended to consider and analyze several other current papers in order to enhance the comprehensiveness of the discussion. Notably, among the 28 references cited, only 9 related to the last 5 years. 

3. The discussion section should encompass an analysis of the innovative contributions introduced by this article, as well as an examination of its limitations. Furthermore, it is imperative to critically examine the limitations of ultrasound, particularly when considering the differential diagnosis of PNSTs in relation to other types of lesions (haemangioma, angioleiomyoma, mixoid neoplasms or tenosynovial giant cell tumors).

4. According to scientific rules and regulations, Table 1 should be shown in the text immediately after its mention, specifically after the first paragraph in the Results section. Additionally, this table provides a brief summary of what will be presented in the results section.

5. Please provide a description and explanation of abbreviations often used in both textual and visual representations (for example CAL from line 183 and others). Including acronyms in the figure description might enhance the accuracy of it.

6. In Figure 6C, I don't find any Doppler signal in the subcutaneous lession.

7. The text from lines 257–259 (Discussion) is similar to lines 93–95 from the introduction. Please revise it!

8. Typographical and linguistic errors are present in the text (for example, lines 46,78, 146, 168,211 and others). A revision of the text is necessary to improve its English language usage.

Comments on the Quality of English Language

Typographical and linguistic errors are present in the text (for example, lines 46,78, 146, 168,211 and others). A revision of the text is necessary to improve its English language usage.

Author Response

The authors would like to thank the Reviewer for their helpful comments and suggestions and for their assistance in improving the quality of the manuscript. The comments are addressed individually below. Changes within the manuscript are highlighted (in red). We believe that our manuscript is improved in the light of the suggested changes.

Dear authors,

I read the study titled "A Retrospective Analysis of High Resolution Ultrasound Evaluation of the 'Split Fat Sign' in Peripheral Nerve Sheath Tumors" with great interest. The main goal of this study was to conduct a retrospective assessment of the presence of heightened accumulation of adipose tissue, specifically referred to as the split fat sign, surrounding benign peripheral nerve sheath tumors (PNSTs) as detected using high-resolution ultrasound.

The pictorial representation shows a remarkable visual appeal, showcasing a limited number of high-quality images. However, in order to meet the criteria for publishing, the article could benefit from further improvements.

  1. Based on the term "split fat sign" and your assertion that this is novel approach, like "no study has described it as an ultrasound feature of PNSTs,", I believe this statement needs to be revised. Several academic papers have been published that examine the alterations associated with adipose tissue growing, alongside the detection PNSTs using ultrasonic imaging. For example, Sonographic Characteristics of Peripheral Nerve Sheath Tumors - David L. Reynolds, Jr., Jon A. Jacobson, Prasuna Inampudi, David A. Jamadar, Farhad S. Ebrahim, and Curtis W. Hayes, American Journal of Roentgenology 2004 182:3, 741-744 or Lawande AD, Warrier SS, Joshi MS. Role of ultrasound in evaluation of peripheral nerves. Indian J Radiol Imaging. 2014 Jul;24(3):254-8. doi: 10.4103/0971-3026.137037. PMID: 25114388; PMCID: PMC4126140.

Thank you for your comment. The sentence has been rewritten and now it reads: “However, few studies have described this sign as a salient ultrasound feature of PNSTs“. Furthermore, the reference of Reynolds et al. (2004), has been included in the introduction section. Now it reads: “The split fat sign described as an increased fat deposition in the upper and lower poles of peripheral nerve sheath tumors, is a commonly described MRI finding, but very few studies have mentioned it as an important ultrasound finding [14,17]

Reynolds DL Jr, Jacobson JA, Inampudi P, Jamadar DA, Ebrahim FS, Hayes CW. Sonographic characteristics of peripheral nerve sheath tumors. AJR Am J Roentgenol. 2004 Mar;182(3):741-4. doi: 10.2214/ajr.182.3.1820741. PMID: 14975979.

  1. From my perspective, the discussions reveal a degree of superficiality. It is recommended to consider and analyze several other current papers in order to enhance the comprehensiveness of the discussion. Notably, among the 28 references cited, only 9 related to the last 5 years.

Thank you for your comment. Several changes have been made to the discussion, and new references have been added to improve the quality of this section. The references added:

Lefebvre G, Le Corroller T. Ultrasound and MR imaging of peripheral nerve tumors: the state of the art. Skeletal Radiol. 2023 Mar;52(3):405-419. doi: 10.1007/s00256-022-04087-5. Epub 2022 Jun 17. PMID: 35713690.

Zheng X, Guo K, Wang H, Li D, Wu Y, Ji Q, Shen Q, Sun T, Xiang J, Zeng W, Chen Y, Wang Z. Extracranial schwannoma in the carotid space: A retrospective review of 91 cases. Head Neck. 2017 Jan;39(1):42-47. doi: 10.1002/hed.24523. Epub 2016 Jul 21. PMID: 27442804.

  1. The discussion section should encompass an analysis of the innovative contributions introduced by this article, as well as an examination of its limitations. Furthermore, it is imperative to critically examine the limitations of ultrasound, particularly when considering the differential diagnosis of PNSTs in relation to other types of lesions (haemangioma, angioleiomyoma, mixoid neoplasms or tenosynovial giant cell tumors).

Thank you for your comment. The discussion section was reviewed and the limitations of the study were added. Now it reads: “Our study has some limitations. First, the sample size was limited and the heterogeneity of the cases analyzed did not allow for comparisons between groups. Furthermore, we did not perform a reproducibility study of the findings, although to avoid the evaluator effect, all examinations were performed by the same expert with more than 10 years’ experience. Finally, it would have been interesting to add a comparison with other diagnostic techniques such as MRI, in order to establish correlations prior to surgery.

  1. According to scientific rules and regulations, Table 1 should be shown in the text immediately after its mention, specifically after the first paragraph in the Results section. Additionally, this table provides a brief summary of what will be presented in the results section.

Done, Table 1 (now Table 2 after reviewer comments) was placed immediately after its mention.

  1. Please provide a description and explanation of abbreviations often used in both textual and visual representations (for example CAL from line 183 and others). Including acronyms in the figure description might enhance the accuracy of it.

Done.

  1. In Figure 6C, I don't find any Doppler signal in the subcutaneous lession.

Thank you for your comment. Figure 6 has been edited to include an image where a doppler signal is found within the affected region.

  1. The text from lines 257–259 (Discussion) is similar to lines 93–95 from the introduction. Please revise it!

Done. Lines 257-259 have been rewritten and now it reads: “The main purpose of this study was to corroborate if the presence of increased deposition of fatty tissue i.e., the split fat sign, around benign PNSTs diagnosed by high resolution ultrasound, was confirmed after surgery, adding a new application of ultrasound in neuroma and schwannoma diagnosis. In addition, we also aimed to corroborate the presence of vascularization around the affected area. Our results showed that up to 90% of cases ultrasound was able to detect fat deposition

  1. Typographical and linguistic errors are present in the text (for example, lines 46,78, 146, 168,211 and others). A revision of the text is necessary to improve its English language usage.

The manuscript grammar and spelling have been checked by Diane Schofield (Chartered Institute of Linguists no. 31814).

Reviewer 5 Report

Comments and Suggestions for Authors

This manuscript reports ultrasound evaluation of the “split fat sign”

in peripheral nerve sheath tumors (PNSTs). The authors state that “no study has described it as an ultrasound feature of PNSTs.” If this is the case, then the manuscript does have some merit. However, the major failing of this manuscript is the lack of quantitative results. Please consider to provide quantitative results to support the conclusion of this work.

1.   The major failing of this manuscript is the lack of quantitative results. Please consider to provide quantitative results to support the conclusion of this work.

2.   A demographic table describing the participants’ demographic information can be provided.

3.   The sampling frequency can be described.

4.   What is the central frequency of the transducer? The manuscript describes a frequency range of “2–11 MHz”. However, it should be noted that if the central frequency is 2 MHz, it cannot be high-frequency, as the title uses “high-frequency”.

5.   Institutional Review Board Statement. Please provide the IRB No. and approval date.

6.   References. Please follow the requirements by the journal.

Comments on the Quality of English Language

Minor improvement can be made.

Author Response

The authors would like to thank the Reviewer for their helpful comments and suggestions and for their assistance in improving the quality of the manuscript. The comments are addressed individually below. Changes within the manuscript are highlighted (in red). We believe that our manuscript is improved in the light of the suggested changes.

This manuscript reports ultrasound evaluation of the “split fat sign” in peripheral nerve sheath tumors (PNSTs). The authors state that “no study has described it as an ultrasound feature of PNSTs.” If this is the case, then the manuscript does have some merit. However, the major failing of this manuscript is the lack of quantitative results. Please consider to provide quantitative results to support the conclusion of this work.

  1. The major failing of this manuscript is the lack of quantitative results. Please consider to provide quantitative results to support the conclusion of this work.

Thank you for your comment. Due to the heterogeneity of the sample, this article is a case series design, so the results are treated individually, without statistical treatment of the data. In addition to the description of the signs found in each of the cases, we have added the total percentage of cases (90%) in which fat accumulation was diagnosed with ultrasound and the finding was corroborated after surgery. Now it reads: “The presence of increased hyperechoic adipose tissue i.e., the split fat sign which is the main focus of our retrospective analysis, was noted in cases of solitary intermuscular and intramuscular peripheral nerve sheath tumors, mainly the schwannomas (90% of cases examined).”

  1. A demographic table describing the participants’ demographic information can be provided.

Thank you for your comment. The demographic data of the participants were included in Table 1. Now it reads:

Table 1. Summary of the demographic data.

PATIENT

Age (years)

Gender

Time evolution (years)

CAL presence

Familiar presence

Case 1

9

Female

9

Yes

Yes

Case 2

45

Male

30

No

Yes

Case 3

38

Female

9

Yes

Yes

Case 4

28

Female

9

Yes

-

Case 5

25

Female

10

Yes

-

Case 6

30

Male

1

No

-

Case 7

32

Male

3

No

-

Case 7

29

Male

0.5

No

-

Case 8

16

Female

4

No

-

Case 9

57

Male

1

No

-

Case 10

40

Male

2

No

-

Case 11

27

Male

1

No

-

       CAL: café-au-lait spots presence.

  1. The sampling frequency can be described.

Thank you for the comment. The sampling frequency was 10Mhz, and it was included in Methods section. Now it reads: “... with a Logiq® P8 ultrasound with a 2–11MHz multifrequency linear probe L3-12-D (the central frequency was 10MHz).”

  1. What is the central frequency of the transducer? The manuscript describes a frequency range of “2–11 MHz”. However, it should be noted that if the central frequency is 2 MHz, it cannot be high-frequency, as the title uses “high-frequency”.

Thank you for your comment. The frequency used in most of the tests was 10Hz. This has been included in the text, and now it reads: “... with a Logiq® P8 ultrasound with a 2–11MHz multifrequency linear probe L3-12-D (the central frequency was 10MHz).“

  1. Institutional Review Board Statement. Please provide the IRB No. and approval date.

Done.

  1. References. Please follow the requirements by the journal.

Done.

Comments on the Quality of English Language

Minor improvement can be made.

The manuscript grammar and spelling have been checked by Diane Schofield (Chartered Institute of Linguists no. 31814). 

Round 2

Reviewer 1 Report

Comments and Suggestions for Authors

No comments

Comments on the Quality of English Language

Minor edits are required.

Author Response

Thank you for your comment. The manuscript grammar and spelling have been checked by Diane Schofield (Chartered Institute of Linguists no. 31814).

Reviewer 3 Report

Comments and Suggestions for Authors

The final version has improved thank you. 

Author Response

The authors would like to thank the Reviewer for their helpful comments and suggestions and for their assistance in improving the quality of the manuscript.

Reviewer 4 Report

Comments and Suggestions for Authors

Dear authors, 

I considered the paper titled "A Retrospective Analysis of High Resolution Ultrasound Evaluation of the 'Split Fat Sign' in Peripheral Nerve Sheath Tumors" to be really intriguing. The primary objective of this study was to retrospectively evaluate the occurrence of increased buildup of fatty tissue, known as the split fat sign, around peripheral nerve sheath tumors (PNSTs) discovered using high-resolution ultrasound.

I appreciate the modifications implemented in the actual version based on the suggestions provided.

Nevertheless, I recommend modifying the title of the article to capitalise each word.

Author Response

The authors would like to thank the Reviewer for their helpful comments and suggestions and for their assistance in improving the quality of the manuscript. 

According to your comments, the title has been modified. Now it reads: "A Retrospective Analysis of High Resolution Ultrasound Evaluation of the 'Split Fat Sign' in Peripheral Nerve Sheath Tumors"

Reviewer 5 Report

Comments and Suggestions for Authors

Thanks for the revision. My concerns have been addressed.

Author Response

(The authors gave the same response as above.)
